# A vaccinia-based single vector construct multi-pathogen vaccine protects against both Zika and chikungunya viruses

Natalie A. Prow[1,2], Liang Liu [3], Eri Nakayama[1,4], Tamara H. Cooper[3], Kexin Yan[1], Preethi Eldi[3], Jessamine E. Hazlewood[1], Bing Tang[1], Thuy T. Le[1], Yin Xiang Setoh[5], Alexander A Khromykh[2,5], Jody Hobson-Peters[5], Kerrilyn R. Diener [3,6], Paul M. Howley[7], John D. Hayball [3,6] & Andreas Suhrbier [1,2]

Zika and chikungunya viruses have caused major epidemics and are transmitted by *Aedes aegypti* and/or *Aedes albopictu*s mosquitoes. The "Sementis Copenhagen Vector" (SCV) system is a recently developed vaccinia-based, multiplication-defective, vaccine vector technology that allows manufacture in modified CHO cells. Herein we describe a single-vector construct SCV vaccine that encodes the structural polyprotein cassettes of both Zika and chikungunya viruses from different loci. A single vaccination of mice induces neutralizing antibodies to both viruses in wild-type and IFNAR$^{-/-}$ mice and protects against (i) chikungunya virus viremia and arthritis in wild-type mice, (ii) Zika virus viremia and fetal/placental infection in female IFNAR$^{-/-}$ mice, and (iii) Zika virus viremia and testes infection and pathology in male IFNAR$^{-/-}$ mice. To our knowledge this represents the first single-vector construct, multi-pathogen vaccine encoding large polyproteins, and offers both simplified manufacturing and formulation, and reduced "shot burden" for these often co-circulating arboviruses.

[1] QIMR Berghofer Medical Research Institute, Brisbane, QLD 4029, Australia. [2] Australian Infectious Disease Research Centre, Brisbane, QLD 4029 and 4072, Australia. [3] Experimental Therapeutics Laboratory, Sansom Institute for Health Research, School of Pharmacy and Medical Sciences, University of South Australia, Adelaide, SA 5000, Australia. [4] Department of Virology I, National Institute of Infectious Diseases, Tokyo 162-8640, Japan. [5] School of Chemistry and Molecular Biosciences, University of Queensland, St Lucia, QLD 4072, Australia. [6] Robinson Research Institute and Adelaide Medical School, University of Adelaide, Adelaide, SA 5005, Australia. [7] Sementis Ltd., Berwick, VIC 3806, Australia. These authors contributed equally: Natalie A. Prow, Liang Liu, Eri Nakayama. Correspondence and requests for materials should be addressed to P.M.H. (email: Paul.Howley@sementis.com.au) or to J.D.H. (email: john.hayball@unisa.edu.au) or to A.S. (email: Andreas.Suhrbier@qimrberghofer.edu.au)

The vaccinia vaccine (VACV) was highly immunogenic and effective at eradicating smallpox globally, and a number of vaccinia-based vaccines and vaccine vectors have subsequently been developed. For instance, a Modified Vaccinia Ankara (MVA) smallpox vaccine, IMVAMUNE®[1], was recently approved in the European Union and Canada. We recently described the Sementis Copenhagen Vector (SCV) vaccine technology and its application to vaccine development[2]. The SCV platform was generated by deleting the *D13L* gene from the Copenhagen strain of VACV. *D13L* encodes an essential viral assembly protein (D13) and its deletion renders SCV incapable of generating viral progeny in vaccine recipients[2]. This approach to attenuation preserves genome amplification, thereby permitting late phase expression of vaccine antigens from the amplified vector genomes[2]. The second unique feature of the SCV vaccines is that it can be produced in a SCV cell substrate (SCS) cell line, which comprises Chinese hamster ovary (CHO) cells stably expressing D13 and CP77. In trans provision of D13 allows SCV assembly and the host-range protein, CP77, imparts VACV and SCV multiplication capability to SCS cells[2]. CHO cells are widely used in the biopharmaceutical industry, with SCV vaccine production in the SCS cell line avoiding many of the issues associated with vaccine manufacture in primary chicken embryo fibroblasts, the cells traditionally used for production of vaccinia-based vaccines[2].

The SCV technology was used to develop a vaccine against chikungunya virus (CHIKV), with the SCV chikungunya vaccine (SCV-CHIK) encoding the structural gene cassette of CHIKV. A

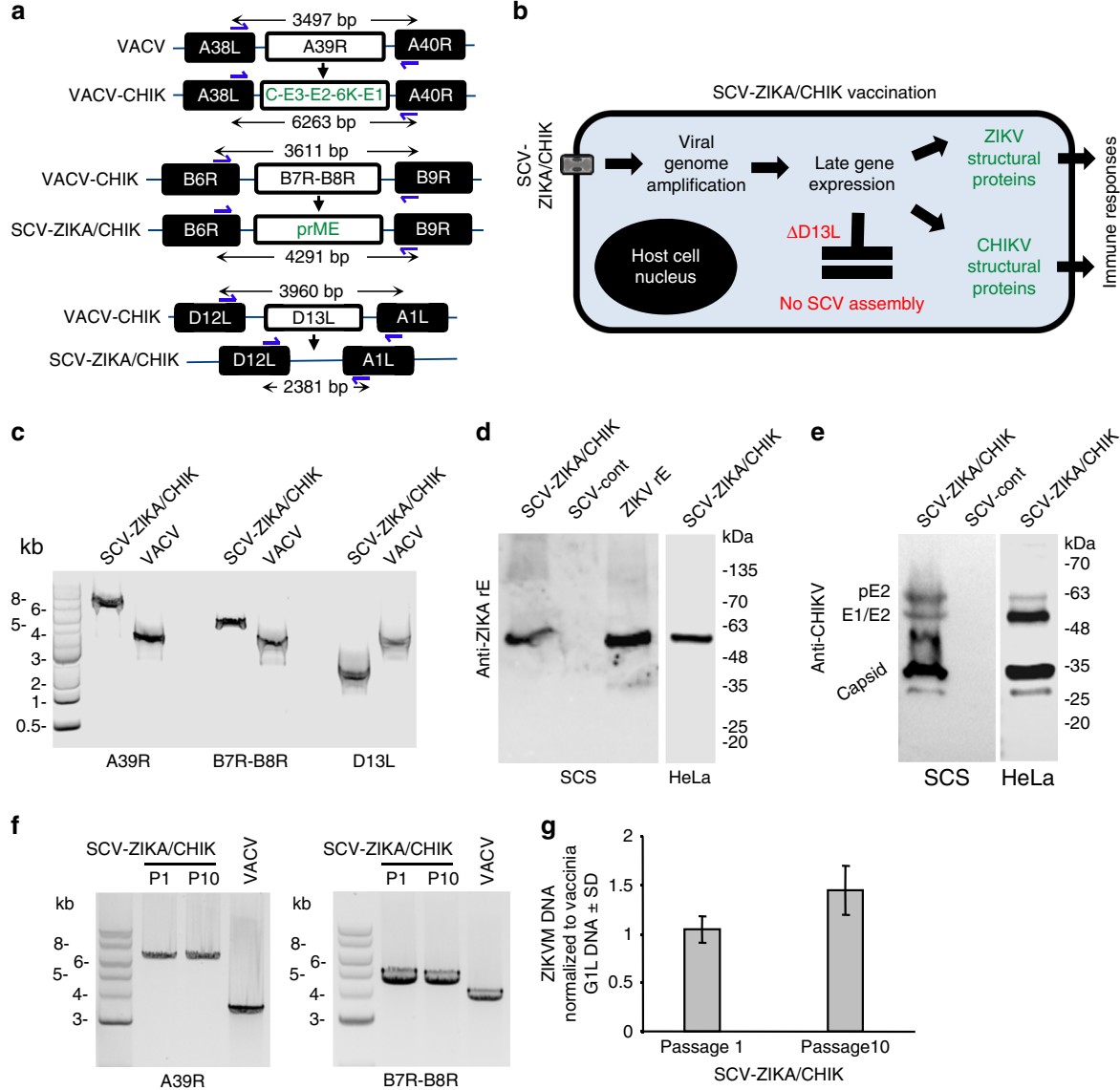

**Fig. 1** SCV-ZIKA/CHIK construction, rationale and characterization. **a** VACV-CHIK was generated from vaccinia virus (VACV) by insertion of the CHIKV structural protein expression cassette inserted into the *A39R* locus. SCV-ZIKA/CHIK was constructed from VACV-CHIK by insertion of ZIKV prME expression cassette into the *B7R-B8R* locus and concurrent deletion of *D13L*. **b** After vaccination the genome is amplified in SCV-ZIKA/CHIK-infected host cells and CHIKV and ZIKV immunogens are expressed from the amplified genomes. Due to the targeted deletion of *D13L*, no viral progeny are generated. **c** PCR of SCV-ZIKA/CHIK and VACV infected SCS cells confirming insertion of *CHIKV* and *ZIKV* genes into *A39R* and *B7R-B8R* loci, respectively, and deletion of *D13L*. **d** Immunoblot of SCV-ZIKA/CHIK and SCV-cont infected SCS and HeLa cells using an anti-ZIKV E antibody, with recombinant ZIKV E (rE) as a positive control. **e** Immunoblot of SCV-ZIKA/CHIK and SCV-cont infected SCS and HeLa cells using a polyclonal anti-CHIKV mouse anti-serum. **f** Lysates of SCV-ZIKA/CHIK-infected cells after one (P1) and ten passages (P10) in SCS cells, were analyzed by PCR (as in **c**) for retention of inserts in the *A39R* and *B7R-B8R* loci. **g** Quantitative PCR of ZIKV M DNA of lysates described in **f**, normalized to VACV G1L DNA. Error bars represent standard deviation ($n = 4$)

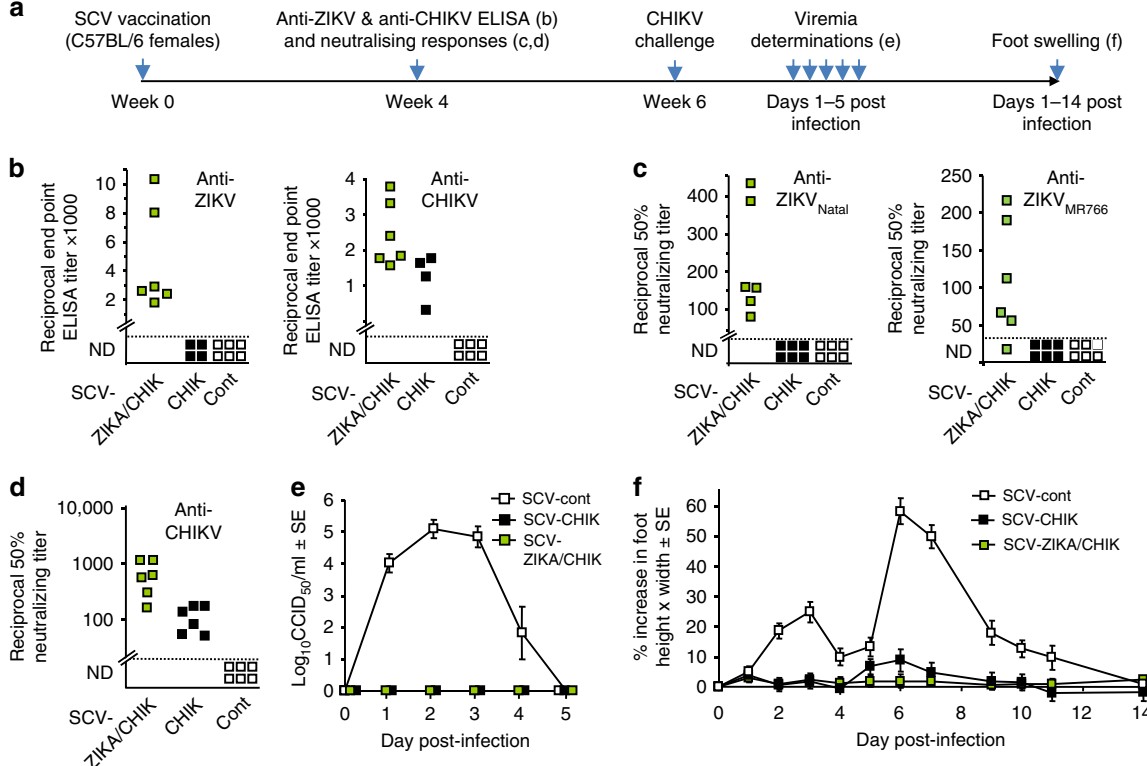

**Fig. 2** ZIKV and CHIKV antibody responses in C57BL/6 mice and CHIKV challenge. **a** Timeline of vaccination, antibody assays, CHIKV challenge, viremia, and foot measurements. **b** End point IgG ELISA titers against ZIKV and CHIKV 4 weeks post vaccination with $10^6$ pfu of the indicated SCV vaccine; SCV-ZIKA/CHIK, SCV-CHIK, or SCV-cont. The limit of detection was 1 in 30, meaning that a 1 in 30 dilution of sera was the highest (starting) concentration of sera used in the assay. ND not detected. ($n = 6$, except for SCV-CHIK $n = 4$ mice per group). SCV-ZIKA/CHIK vaccinated mice had significantly higher CHIKV and ZIKV titers than SCV-cont vaccinated mice (both $p = 0.005$, Kolmogorov–Smirnov tests). (Differences in anti-CHIKV titers between SCV-CHIK and SCV-ZIKA/CHIK were not significant). **c** Neutralizing titers against $ZIKV_{Natal}$ and $ZIKV_{MR766}$ in mice vaccinated with the indicated SCV vaccines ($n = 6$ per group). Limit of detection 1 in 30. ($p = 0.005$ and $0.031$, Kolmogorov–Smirnov tests). **d** Neutralizing titers against CHIKV. Limit of detection 1 in 30. (Compared with SCV-cont both $p = 0.005$, Kolmogorov–Smirnov tests). **e** Viremia of mice described in d, after challenge with CHIKV (6 weeks post vaccination). Limit of detection 2 $log_{10}CCID_{50}$/ml. For days 1–3 the viremia in SCV-ZIKA/CHIK vaccinated mice was significantly lower than in SCV-control vaccinated mice; all $p = 0.005$, Kolmogorov–Smirnov tests. **f** Foot swelling of mice described in e. From days 2–10 the foot swelling in SCV-ZIKA/CHIK vaccinated mice was significantly lower than in SCV-control vaccinated mice; $p = 0.03$–$0.001$, Mann–Whitney $U$ tests. Error bars represent standard error of the mean

single vaccination with SCV-CHIK provided protection against CHIKV infection and arthritic diseases in a wild-type (C57BL/6) adult mouse model[2], which recapitulates many aspects of human CHIKV disease[3,4]. CHIKV is a mosquito-borne alphavirus that is primarily associated with acute and chronic rheumatic symptoms, with occasional infections also resulting in severe disease manifestations and mortality[5,6]. Hospitalization rates for CHIKV patients range from ≈2.3 to ≈13%, with a 5 ± 7 day mean length of stay (in Reunion Island)[7]. Although CHIKV has in the past been associated with sporadic outbreaks around the world, in 2004 CHIKV re-emerged to produce the largest epidemic ever recorded for this virus, with millions of cases reported globally, primarily in Africa, Asia, and South and Central America[5,6]. Autochthonous transmission has also occurred in Europe and the USA. The primary mosquito vector species for CHIKV are *Aedes aegypti* and *Aedes albopictus*[8], both are highly invasive and, due to human activity, have attained global distributions[9].

Zika virus (ZIKV) (family *Flaviviridae*) represents another mosquito-borne virus that has recently caused global health concerns due to its association with congenital Zika syndrome (CZS). CZS encompasses a spectrum of predominantly neurological complications (including but not limited to microcephaly) arising from infection of fetal brains[10,11]. Although infected pregnant women often have no or only mild symptoms, the virus

appears able to cross the placenta[12,13] and infect and destroy (primarily) neural progenitor cells in the fetal central nervous system[14]. Although CZS has now been well documented for the outbreak in Brazil, there remains a question of whether similar manifestations have gone unnoticed in Africa[15]. ZIKV is also associated with Guillain-Barré syndrome. ZIKV is primarily transmitted by *Aedes aegypti* mosquitoes[16]. Some data suggests circulation of ZIKV in wild *Aedes albopictus* populations[17], with laboratory vector competence studies also suggesting ZIKV can be transmitted by this mosquito species[18]. Sexual transmission of ZIKV has also been documented, with the virus able to infect, damage and persist in testes[19].

ZIKV and CHIKV co-circulate in many parts of the world[20], with human co-infections reported in several countries[21–24] and *Aedes aegypti* mosquitoes are also able to co-transmit both viruses[25].

Herein we described a SCV multi-pathogen vaccine[26] where a single-vector construct encodes (from different loci) the complete structural polyprotein cassettes of both ZIKV (prME, 2016 nucleotides) and CHIKV (C-E3-E2-6K-E1, 3747 nucleotides). This SCV-ZIKA/CHIK vaccine provided protection against CHIKV infection and disease in the adult CHIKV mouse model. The vaccine also induced anti-ZIKV antibody responses in C57BL/6 and IFNAR$^{-/-}$ mice, and provided protection in three

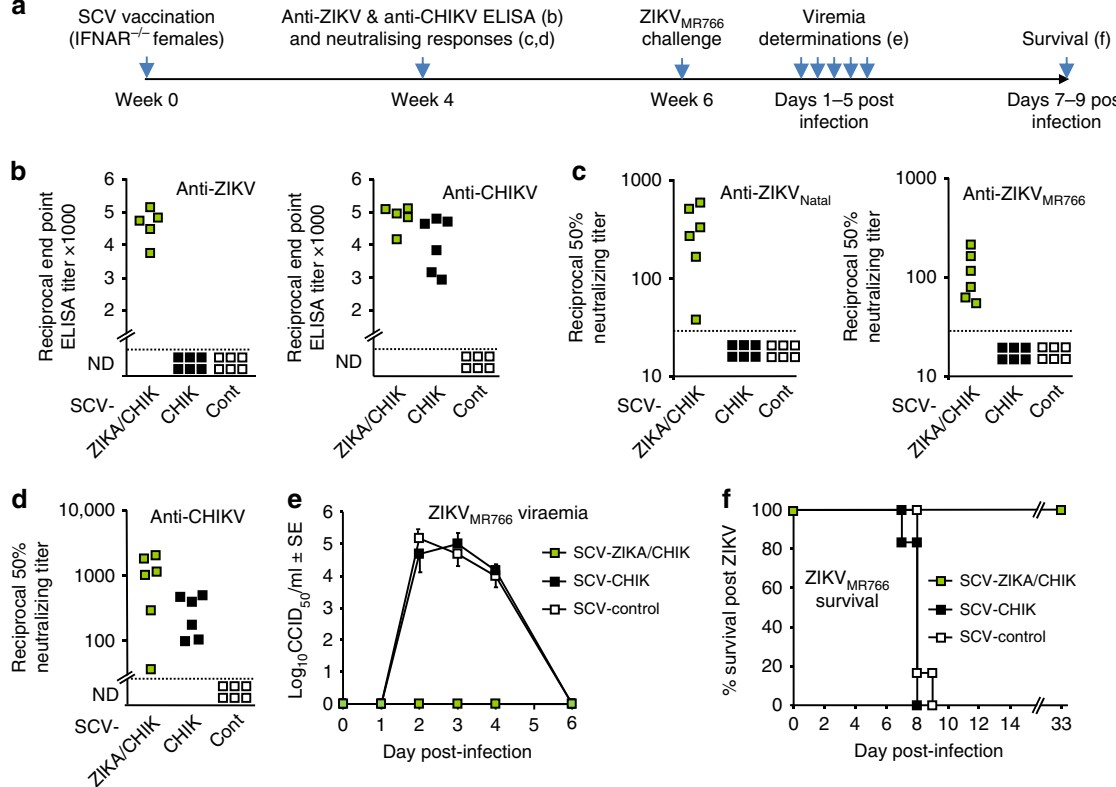

**Fig. 3** ZIKV and CHIKV antibody responses in IFNAR$^{-/-}$ female mice and ZIKV$_{MR766}$ challenge. **a** Timeline of vaccination, antibody assays, ZIKV challenge, viremia and survival determinations. **b** End point IgG ELISA titers against ZIKV and CHIKV 4 weeks post vaccination with $10^6$ pfu of the indicated SCV vaccine. Limit of detection 1 in 30 dilution; ND not detected ($n = 5/6$ mice per group). SCV-ZIKA/CHIK vaccinated mice had higher ZIKV and CHIKV titers than SCV-control vaccinated mice (all $p = 0.009$, Kolmogorov–Smirnov tests). (Differences in anti-CHIKV titers between SCV-CHIK and SCV-ZIKA/CHIK were not significant). **c** Neutralization titers against ZIKV$_{Natal}$ and ZIKV$_{MR766}$ in mice vaccinated with the indicated SCV vaccines ($n = 6$ per group). (Verses SCV-cont, both $p = 0.005$, Kolmogorov–Smirnov tests). **d** Neutralization titers against CHIKV in mice vaccinated with the indicated SCV vaccines ($n = 6$ per group). (Verses SCV-cont, both $p = 0.005$, Kolmogorov–Smirnov tests). (Differences in anti-CHIKV titers between SCV-CHIK and SCV-ZIKA/CHIK were not significant). **e** Viremia after challenge with ZIKV$_{MR766}$ (6 weeks post vaccination). Limit of detection 2 $\log_{10}CCID_{50}$/ml. For days 2–4 the viremia in SCV-ZIKA/CHIK vaccinated mice ($n = 5/6$ per group) was significantly lower than in SCV-control vaccinated mice (all $p = 0.009$, Kolmogorov–Smirnov tests). Error bars represent standard error of the mean. **f** Survival of mice described in e. Mice were euthanized when ethically defined end points had been reached. SCV-ZIKA/CHIK vaccinated mice survived significantly longer than SCV-control vaccinated mice ($p = 0.001$, log rank, Mantel–Cox, test)

ZIKV models: (i) survival in IFNAR$^{-/-}$ mice infected with the mouse-adapted prototype African ZIKV$_{MR766}$ strain of ZIKV, which is usually lethal in these GMO animals[27]; (ii) fetal outcomes in IFNAR$^{-/-}$ dams infected with ZIKV$_{Natal}$, an unpassaged ZIKV isolate unequivocally associated with microcephaly[27]; and (iii) testes infection and pathology in IFNAR$^{-/-}$ males infected with ZIKV$_{Natal}$.

## Results

**Design rationale for the SCV-ZIKA/CHIK vaccine**. The Copenhagen strain of VACV (VACV) was originally used as a smallpox vaccine in Denmark and the Netherlands and provides the source virus for SCV vaccines[2]. To construct SCV-ZIKA/CHIK, VACV-CHIK was first generated by replacing the *A39R* gene of VACV with the structural protein cassette of CHIKV (Capsid-E3-E2-6K-E1) to generate VACV-CHIK (Fig. 1a) as described[2]. The *B7R-B8R* genes of VACV-CHIK were then replaced with the structural protein cassette of ZIKV (prME) (Fig. 1a). (*B8R* encodes a secreted interferon-γ receptor homolog, with its deletion from VACV previously shown to attenuate the virus, whilst immunogenicity was retained[28]. *B7R* encodes a protein found in the endoplasmic reticulum, whose deletion in VACV resulted in smaller skin lesions in mice[29]). The

multiplication-defective SCV-ZIKA/CHIK was ultimately generated by concurrent deletion of the *D13L* gene[2] (Fig. 1a).

After vaccination, SCV-ZIKA/CHIK infects host cells and the SCV genome (which encodes the ZIKV and CHIKV immunogens) is amplified (to about ≈10,000 copies[30,31]) (Fig. 1b). No infectious progeny can be generated due to the absence of the assembly protein, D13[2] (Fig. 1b). Late gene expression from the amplified viral genomes then results in expression of the ZIKV and CHIKV structural protein immunogens (Fig. 1b).

**Characterization of the SCV-ZIKA/CHIK vaccine construct**. The insertion of the *CHIKV* genes into the *A39R* locus, the insertion of the *ZIKV* genes into the *B7R-B8R* locus, and the deletion of *D13L* were confirmed by PCR of SCV-ZIKA/CHIK-infected SCS cells (Fig. 1c). Lysates of SCV-ZIKA/CHIK-infected SCS and HeLa cells were also analyzed by immunoblotting. This illustrated expression of authentically processed, polyprotein-derived ZIKV E (Fig. 1d) and CHIKV structural proteins (Fig. 1e) in SCS cells (*D13L*-expressing cells in which SCV-ZIKA/CHIK can generate viral progeny) and in human-derived HeLa cells (*D13L*-negative cells in which SCV-ZIKA/CHIK cannot generate viral progeny). Furthermore, CHIKV structural protein expression

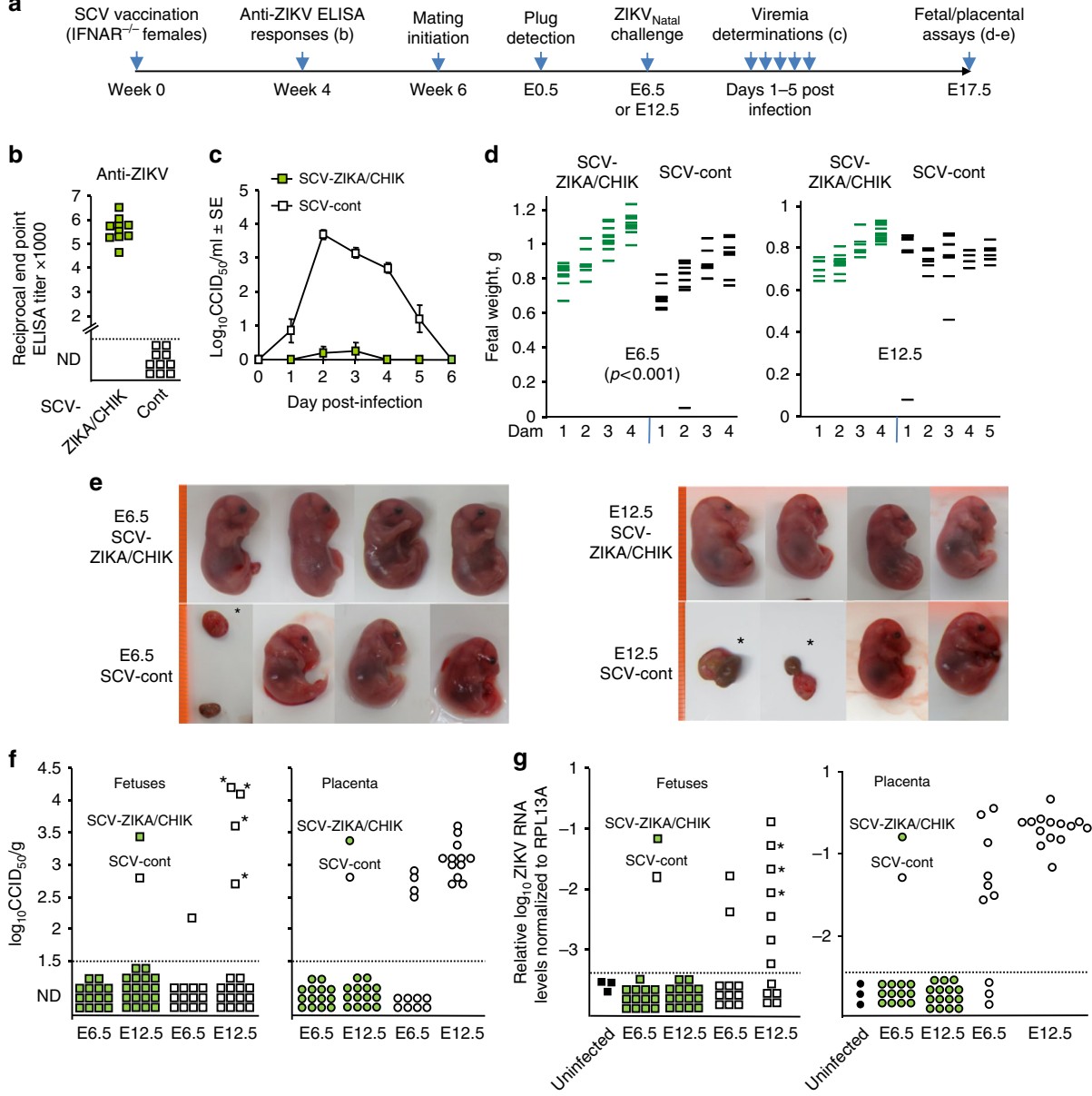

**Fig. 4** Challenge of vaccinated pregnant IFNAR$^{-/-}$ dams with ZIKV$_{Natal}$. **a** Timeline of vaccination, antibody responses, mating, challenge, viremia, and fetal/placental assays. **b** Anti-ZIKV serum IgG ELISA titers 4 weeks after vaccination with SCV-ZIKA/CHIK or SCV-cont; ($p < 0.001$ Kolmogorov–Smirnov tests). **c** Viremia in the vaccinated mice ($n = 10$ per group). (For days 2–4 all $p < 0.001$, Kolmogorov–Smirnov tests). Error bars represent standard error of the mean. **d** Fetal weights at E17.5 for SCV-vaccinated dams ($n = 4$–5 dams per group) infected with ZIKV$_{Natal}$ either at E6.5 or E12.5 (for fetuses $n = 36$, 33, 35, 43 left to right). SCV-cont vaccinated dams had a lower mean fetal weight than SCV-ZIKA/CHIK vaccinated dams when challenged at E6.5 ($p < 0.001$, $t$ test, $n = 36$ and 33). Fetal/placental masses were not included in this graph. **e** Photographs of selected fetuses (E17.5). Highly deformed fetuses and placenta, and fetal/placental masses are indicated by asterisks (*). The orange lines on the left represent a ruler with 1 mm marks. **f** Fetal and placenta ZIKV tissue titers at E17.5 from SCV-vaccinated dams (at least three fetal/placental tissues from each of the 4/5 litters were tested); fetal heads or placenta, except * indicating deformed fetal/placental masses. Limit of detection was 1.5 log$_{10}$CCID$_{50}$/g. The positive placenta in SCV-cont E6.5 group were derived from three litters. Titers were significantly lower in the SCV-ZIKA/CHIK vaccinated groups compared to the SCV-cont groups $p < 0.001$ (Kolmogorov–Smirnov test) combining data from F and Pl, and E6.5 and E12.5. **g** qRT PCR of fetal heads. Three uninfected fetal heads (black squares) were analyzed in triplicate and the highest value plus 3 SD was used as a cutoff (all other data points below the dashed cutoff line are not plotted to scale). *Deformed. Statistics as for **f**, $p = 0.001$ combining data from E6.5 and E12.5

in SCV-ZIKA/CHIK-infected HeLa cells was not lower than in SCV-CHIK[2] infected HeLa cells (Supplementary Fig. 1).

**SCV-ZIKA/CHIK immunogen insert stability**. To assess the stability of the immunogen inserts, SCV-ZIKA/CHIK was passaged 10 times in SCS cells (MOI = 0.01–0.001, 3 day culture per passage), with virus from passage 1 and passage 10 analyzed by PCR (Fig. 1f) and quantitative PCR (Fig. 1g). The correct size of (i) the CHIKV structural protein insert into the A39R locus, and (ii) the ZIKV prME insert into the B7R–B8R locus, was retained after 10 passages, with no evidence of deletions (Fig. 1f). In addition, quantitative PCR of ZIKV M protein DNA, normalized to the *G1L* gene of VACV[32], showed no reduction in levels of M protein DNA over 10 passages.

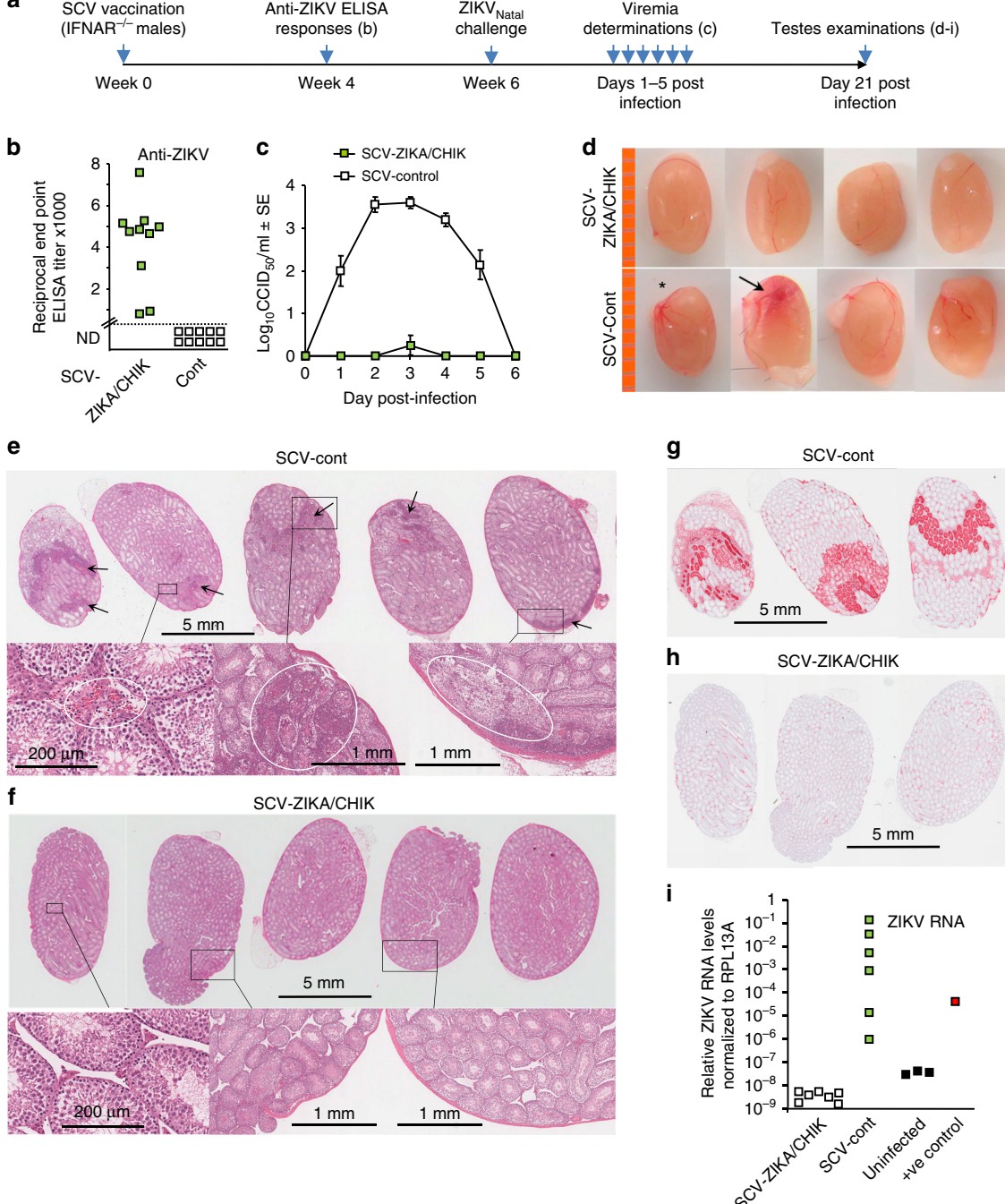

**Fig. 5** Challenge of vaccinated male IFNAR$^{-/-}$ mice with ZIKV$_{Natal}$. **a** Timeline of SCV vaccination, challenge, viremia and testes examinations. **b** Anti-ZIKV serum IgG ELISA titers 4 weeks after vaccination with SCV-ZIKA/CHIK or SCV-cont ($p < 0.001$, Kolmogorov–Smirnov test). **c** Viremia in SCV-vaccinated mice after challenge with ZIKV$_{Natal}$ ($n = 10$ mice per group) (days 2–4 all $p < 0.001$, Kolmogorov–Smirnov tests). Error bars represent standard error of the mean. **d** Pictures of testes taken day 21 after challenge. *One testes (out of 12) was slightly smaller, another showed signs of hemorrhage (arrow). The orange line on the left represents a ruler with 1 mm marks. **e** H&E staining of testes from SCV-cont vaccinated mice showing dark patches of cellular infiltrates (arrows); top row bar = 5 mm. Enlargements of selected areas show (in white circles, left to right) hemorrhage around Leydig cells (in the same testes arrowed in **d**), high density of inflammatory infiltrates in and around the seminiferous tubules, and destruction of seminiferous tubules; bottom row, bar = 200 μm (left) and bars = 1 mm (2 right hand images). **f** H&E staining of testes from SCV-ZIKA/CHIK vaccinated mice. Enlargements of selected areas show normal testis architecture with no discernible lesions. Bars as in **e**. **g** Immunohistochemistry with 4G4 (anti-NS1 monoclonal antibody) of serial sections of the three testes shown top left in **e**. Bar = 5 mm. **h** Immunohistochemistry with 4G4 of serial sections of the three testes shown top left in **f**. Bar = 5 mm. **i** ZIKV RNA qRT PCR of testes day 21 post infection. SCV-ZIKA/CHIK vs SCV-cont, $p = 0.003$ (Kolmogorov–Smirnov test); ($n = 6/7$ testes from 6/7 mice per SCV vaccine group). Three uninfected negative control testes from three mice and one known positive control sample were included

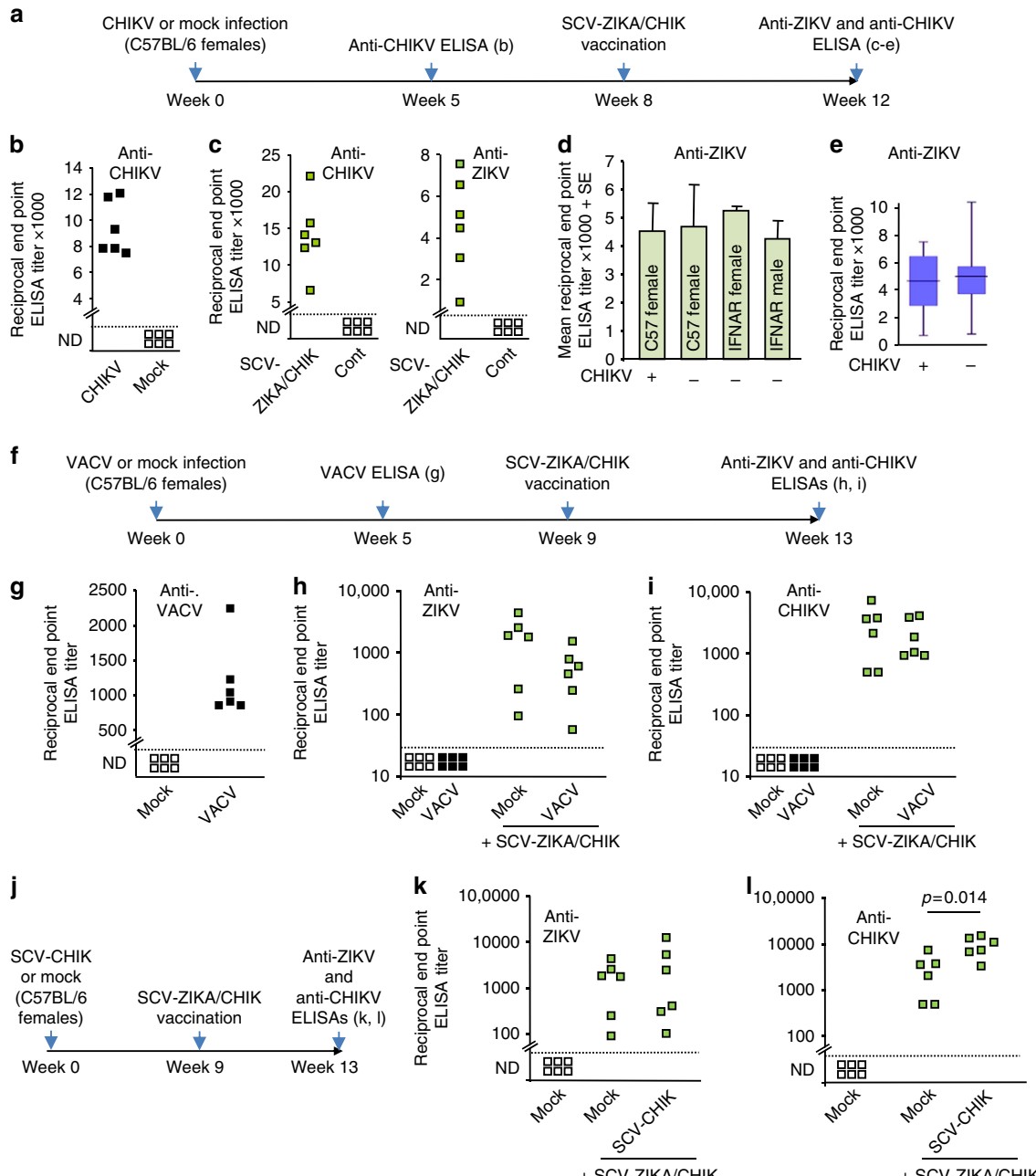

**Fig. 6** Effects of prior CHIKV or VACV infections on SCV-ZIKA/CHIK vaccination. **a** Timeline of CHIKV infection, SCV vaccination and determination of antibody responses to ascertain whether prior CHIKV infection affects anti-ZIKV responses after SCV-ZIKA/CHIK vaccination. **b** Mice were infected with CHIKV or were mock infected with PBS and anti-CHIKV antibody titers determined 5 weeks post infection by ELISA ($n = 6$ mice per group). **c** Mice that had been infected with CHIKV were vaccinated with SCV-ZIKA/CHIK and after 4 weeks anti-CHIKV and anti-ZIKV serum ELISA titers were determined. Sera from mice vaccinated with SCV-cont were included in the assays as negative controls. **d** Mean anti-ZIKV serum ELISA titers 4 weeks after SCV-ZIKA/CHIK vaccination in the indicated groups ($n = 6, 6, 15, 10$, left to right), either with prior CHIKV infection (+) or without (−). Error bars represent standard error of the mean. **e** Box and whiskers plots using the same data from **d** to compare anti-ZIKV ELISA titers obtained in mice that had previously been infected with CHIKV (+) ($n = 6$) with those that had not (−) ($n = 31$). Box—upper and lower quartile, with bar as median. Whiskers—maximum and minimum values. **f** Timeline of VACV infection, SCV vaccination and determination of antibody responses to ascertain whether prior VACV infection affects anti-ZIKV and anti-CHIKV responses after SCV-ZIKA/CHIK vaccination. **g** VACV responses after VACV infection or mock infection (PBS). **h** Anti-ZIKV ELISA titers obtained in mice that had previously been mock infected or infected with VACV. Also shown are responses after SCV-ZIKA/CHIK vaccination in mice that had previously received a VACV or mock infection. Limit of detection 1 in 30 dilution. **i** As for **h** except measuring anti-CHIKV ELISA titers. **j** Timeline of vaccinations and ELISAs to ascertain whether prior SCV-CHIK vaccination affects anti-ZIKV and anti-CHIKV responses after SCV-ZIKA/CHIK vaccination. **k** Anti-ZIKV ELISA titers obtained in mice that had previously been mock infected or vaccinated with SCV-CHIK and then vaccinated with SCV-ZIKA/CHIK. **l** Anti-CHIKV ELISA titers obtained in mice that had previously been mock infected or vaccinated with SCV-CHIK and then vaccinated with SCV-ZIKA/CHIK (statistics by $t$ test)

**Vaccination of C57BL/6 mice and CHIKV challenge**. The timeline of vaccination, challenge and analyses is shown in Fig. 2a. SCV-ZIKA/CHIK, SCV-CHIK[2], and SCV-cont (control SCV vector encoding DsRed[2]) were produced in SCS cells. The SCV vaccines were used to vaccinate C57BL/6 mice once ($10^6$ pfu) by the intramuscular route (Fig. 2a, week 0). Four weeks after vaccination, anti-ZIKV and anti-CHIKV IgG titers were determined by ELISA. SCV-ZIKA/CHIK, but not SCV-CHIK (or SCV-cont), induced significant antibody responses to ZIKV (Fig. 2b, anti-ZIKV), whereas both SCV-ZIKA/CHIK and SCV-CHIK (but not SCV-cont) induced significant responses to CHIKV (Fig. 2b, anti-CHIKV). SCV-ZIKA/CHIK, but not SCV-CHIK (or SCV-cont), also induced significant neutralizing responses to both $ZIKV_{Natal}$ and $ZIKV_{MR766}$ (Fig. 2c), with both SCV-ZIKA/CHIK and SCV-CHIK inducing neutralizing antibody responses to CHIKV (Fig. 2d).

Vaccinated mice were challenged with CHIKV 6 weeks after vaccination (Fig. 2a, week 6). Both SCV-ZIKA/CHIK and SCV-CHIK vaccinated mice were protected against the development of a detectable CHIKV viremia (Fig. 2e) and the ensuing foot swelling (Fig. 2f), which is a measure of CHIKV-induced arthritis[4].

**Vaccination of IFNAR$^{-/-}$ mice and ZIKV$_{MR766}$ challenge**. The timeline of vaccination, challenge and analyses is shown in Fig. 3a. Significant anti-ZIKV ELISA titers were seen after a single SCV-ZIKA/CHIK vaccination of IFNAR$^{-/-}$ mice (Fig. 3b, anti-ZIKV) and significant anti-CHIK ELISA titers were seen after SCV-ZIKA/CHIK and SCV-CHIK vaccination (Fig. 3b, anti-CHIKV). Significant anti-ZIKV$_{Natal}$ and ZIKV$_{MR766}$ neutralizing titers were also seen after SCV-ZIKA/CHIK vaccination (Fig. 3c), and significant anti-CHIKV neutralizing titers were seen after SCV-ZIKA/CHIK and SCV-CHIK vaccination (Fig. 3d).

After challenge with ZIKV$_{MR766}$, SCV-ZIKA/CHIK vaccinated mice showed no detectable viremia (Fig. 3e) and showed 100% protection against mortality, whereas SCV-CHIK and SCV-cont vaccinated mice developed severe disease and were killed (Fig. 3f). SCV-ZIKA/CHIK vaccination thus provided complete protection against challenge with the generally lethal ZIKV$_{MR766}$ strain of ZIKV[27].

**Vaccination of IFNAR$^{-/-}$ dams and ZIKV$_{Natal}$ challenge**. The timeline of vaccination, challenge and analyses is shown in Fig. 4a. Following a single SCV-ZIKA/CHIK vaccination, significant anti-ZIKV ELISA titers were seen in female IFNAR$^{-/-}$ mice (Fig. 4b). Mice were then mated and after plug detection were weighed; if weight increased by >1 g at E6.5 (confirming pregnancy), dams were challenged with ZIKV$_{Natal}$ (at E6.5 or E12.5, nominally representing early and mid gestation, respectively). Viremia was significantly and substantially suppressed in SCV-ZIKA/CHIK vaccinated dams compared to SCV-cont vaccinated dams (Fig. 4c). ZIKV$_{Natal}$ infection was previously reported to be asymptomatic in IFNAR$^{-/-}$ female mice >8 weeks old[27], and no symptoms were seen in infected dams in the current study.

At E17.5, dams were euthanized and fetal weights determined. SCV-ZIKA/CHIK vaccinated dams infected at E6.5 had fetuses with significantly higher weights at E17.5 when compared to SCV-cont vaccinated dams (t test, $p < 0.001$) (Fig. 4d, E6.5). Fetal weights were on average 17.3 % lower (0.8 ± SE 0.032 g, $n = 33$) in SCV-cont vaccinated dams when compared to SCV-ZIKA/CHIK vaccinated dams (0.97 ± SE 0.022 g, $n = 36$). (In humans birth weights are also often lower for neonates from ZIKV infected mothers[33]). The weight difference in the murine fetuses was not due to litter size differences as litter sizes were actually slightly

larger in the SCV-ZIKA/CHIK vaccinated dams (8, 9, 9, and 10) compared with the SCV-cont vaccinated dams (7, 7, 9, 10) (Fig. 4d, E6.5, left to right). Fetal weights were not significantly different when dams were infected at E12.5; (litter sizes where 8, 11, 10, 10 and 8, 10, 11, 11, 9, Fig. 4d, E12.5 left to right).

Photographs of selected fetuses are shown in Fig. 4e, with the deformed fetuses and fetal/placental masses in the SCV-cont vaccinated animals indicated (Fig. 4e, *). Fetuses, placenta and fetal/placental masses were either analyzed for viral titers (Fig. 4f) or where subject to qRT PCR (Fig. 4g), with no virus or viral RNA detected in tissues from SCV-ZIKA/CHIK vaccinated mice.

**Vaccination of IFNAR$^{-/-}$ males and ZIKV$_{Natal}$ challenge**. The timeline of vaccination, challenge and analyses are shown in Fig. 5a. Following a single vaccination of IFNAR$^{-/-}$ male mice with SCV-ZIKA/CHIK, significant anti-ZIKV ELISA antibody responses were generated (Fig. 5b). SCV-ZIKA/CHIK vaccinated mice also showed a significant and substantial reduction in viremia after challenged with ZIKV$_{Natal}$ (Fig. 5c). On day 21 after challenge mice were euthanized and testes removed. Testes from SCV-cont vaccinated mice were overtly normal except one testis (out of 12, $n = 6$ mice per group) was slightly smaller (Fig. 5d, *) and another showed signs of hemorrhage (Fig. 5d, arrow).

Histology of testes from SCV-cont vaccinated mice ($n = 6$ mice, 1 testis from each mouse) showed lesions (hemorrhage, infiltrates and/or destruction of seminiferous tubules[34]) in 5/6 testes (from six mice) (Fig. 5e). In contrast, none of the testes from SCV-ZIKA/CHIK vaccinated mice ($n = 7$ mice, 1 testis from each mouse) showed any discernible lesions (Fig. 5f). Immunohistochemistry with 4G4 (pan-flavivirus anti-NS1 antibody[27]) illustrated strong staining in and around the lesions, with staining primarily localizing to seminiferous tubules (Fig. 5g; serial sections of the three testes shown top left in e). In contrast, testes from SCV-ZIKA/CHIK vaccinated mice showed minimal staining (Fig. 5h; serial sections of the three testes shown top left in f).

qRT PCR of the testes showed a significantly lower ($p = 0.003$, Kolmogorov–Smirnov test) level of ZIKA RNA in the testes from SCV-ZIKA/CHIK vaccinated mice compared with testes from SCV-cont mice (Fig. 5i). Levels in testes from SCV-ZIKA/CHIK vaccinated mice were no higher than those in uninfected mice (Fig. 5i), suggesting complete clearance of ZIKV RNA in testes from the former animals (mice were killed on day 21 after infection and for each SCV-vaccinated mouse, one testis was used for histology and the other for qRT PCR).

**SCV-ZIKA/CHIK vaccination after CHIKV or VACV or SCV-CHIK**. One might speculate that the ability of SCV-ZIKA/CHIK vaccination to induce anti-ZIKV responses might be compromised in individuals that have anti-CHIKV-immune responses due to a prior exposure to CHIKV[35,36]. To test this contention, C57/BL6 mice were infected with CHIKV and were then vaccinated with SCV-ZIKA/CHIK (Fig. 6a). After CHIKV infection, mice showed robust anti-CHIKV ELISA IgG responses (Fig. 6b), and after SCV-ZIKA/CHIK vaccination these responses increased (Fig. 6c, anti-CHIKV). Importantly, in CHIKV-immune mice, SCV-ZIKA/CHIK vaccination efficiently generated anti-ZIKV responses (Fig. 6c, anti-ZIKV). These latter responses were not significantly different from those obtained in mice with no prior CHIKV infection, when comparing either means (Fig. 6d) or medians (Fig. 6e). Anti-ZIKV antibody induction by SCV-ZIKA/CHIK was thus not compromised by pre-existing anti-CHIKV immunity.

Anti-vector immunity can compromise the ability of a recombinant vaccine vector effectively to induce immune

responses. To test this for SCV, mice were either mock infected or infected with VACV (Fig. 6f), with the latter mice shown to have generated anti-VACV antibody responses (Fig. 6g). Mice were then vaccinated with SCV-ZIKA/CHIK and anti-ZIKV and anti-CHIKV ELISA titers determined (Fig. 6f). Neither anti-ZIKV (Fig. 6h) nor anti-CHIKV antibody responses (Fig. 6i) were significantly affected by prior VACV infection. These results are consistent with studies using recombinant MVA vaccines[37,38] and argue that pre-existing anti-VACV immunity does not effectively suppress the ability of these poxvirus vaccines to induce antibody responses to recombinant immunogens. Prior vaccination with SCV-CHIK also did not significantly affect anti-ZIKV responses following SCV-ZIKA/CHIK vaccination (Fig. 6j,k), although anti-CHIKV responses were significantly boosted (Fig. 6l).

## Discussion

Herein we describe the application of the SCV vaccine technology to the development of a single vector, multi-pathogen, SCV-ZIKA/CHIK vaccine. The vaccine was able to prevent viremia and arthritic disease in a CHIKV wild-type mouse model and to mediate protection in three models of ZIKV infection using IFNAR$^{-/-}$ mice. A single vaccination protected against (i) lethal infection with ZIKV$_{MR766}$, (ii) fetal/placental ZIKV$_{Natal}$ infection in pregnant IFNAR$^{-/-}$ dams, and (iii) ZIKV$_{Natal}$ infection and testes damage in male IFNAR$^{-/-}$ mice.

Vaccinia-based vaccines have a number of features that make them attractive for managing epidemics in resource poor settings, primarily cold chain-independent distribution capacity[39] and long lasting immunity[40]. The inability of SCV vaccines to generate viral progeny in human cells or to cause disease in immunodeficient mice[2] suggests SCV will have a similar safety profile as the replication-deficient, passage-attenuated MVA. MVA has an impeccable safety record, with more than 120,000 people vaccinated at the end of the smallpox eradication campaign without any serious adverse effects[41]. In addition, MVA was recently shown to be safe and well tolerated in individuals with atopic dermatitis or HIV[42,43]. The CHO-based SCS line also provides a rapid production and scale up capacity for SCV vaccine manufacture[2].

Prior exposure to CHIKV did not significantly affect the ability of SCV-ZIKA/CHIK to induce antibody responses to ZIKV. Concepts related to "original antigenic sin"[35] might raise concerns that pre-existing immune responses to CHIKV could inhibit induction of anti-ZIKV responses. However, the ZIKV and CHIKV polyproteins are expressed from separate loci in the SCV-ZIKA/CHIK vaccine, with these two immunogens not physically linked, a prerequisite for the classically described "sin" phenomenon[35]. Concerns that pre-existing anti-CHIKV cytotoxic T cell (CTL) responses[44] might kill SCV-ZIKA/CHIK-infected cells before sufficient ZIKV antigens can be made, might be ameliorated by the recent insight that CTL killing activity in vivo is actually rather limited[45], with poxviruses also deploying a plethora of strategies to inhibit apoptosis[46]. Prior infection with VACV had no significant impact on anti-ZIKV and anti-CHIKV antibody induction after SCV-ZIKA/CHIK vaccination, an observation consistent with studies on recombinant MVA vaccines[37,38]. Prior SCV-CHIK vaccination also had no significant impact on anti-ZIKV antibody induction after SCV-ZIKA/CHIK vaccination. Vaccinia deploys a series of strategies to avoid antibody neutralization[47], and human studies have illustrated that although anti-VACV immunity effectively inhibits VACV dissemination, it is much less effective at inhibiting initial infection by VACV[48]. SCV vaccines do not disseminate in vivo[2], with a single round infection sufficient to induce immunity.

When comparing the anti-CHIKV responses after SCV-ZIKA/CHIK and SCV-CHIK, no consistent significant difference was observed (for Figs. 2d and 3b,d, $p = 0.06–0.3$, Kolmogorov–Smirnov tests). Furthermore, anti-ZIKV antibody responses after SCV-ZIKA/CHIK and SCV-ZIKA vaccination were also not significantly different (Supplementary Fig. 2). Interference or interaction effects between the ZIKV and CHIKV immunogens thus appear to be minimal in this context.

To the best of our knowledge this represents the first efficacious single vector, multi-pathogen vaccine that encodes whole structural polyprotein gene cassettes from two unrelated viruses. The large payload capacity of poxvirus vectors (at least 25,000 bp[49]) is relatively unique, with most established virus vector systems unable to package genomes containing such large recombinant inserts. An immunogenic multi-pathogen VACV construct has previously been reported and encoded (from three insertion sites) single antigens from influenza, hepatitis B, and herpes simplex virus[50]. A recently reported multi-pathogen MVA vaccine, encoding hepatitis B and rabies virus immunogens under the control of a T7 promoter, required co-infection with a MVA encoding T7 polymerase, but failed to induce responses to the heterologous immunogens[51]. A trivalent MVA vaccine encoding three H5N1 influenza hemagglutinin genes was efficacious and used three different promoters and a single-insertion site[52]. Herein we used the poxvirus synthetic strong early late promoter[53]; one to drive CHIKV, and another to drive ZIKV, polyprotein expression. Despite the potential for homologous recombination, the SCV-ZIKA/CHIK construct remained stable over 10 passages. As these two identical 39 nucleotide long promoters are widely separated in the SCV genome, any intramolecular homologous recombination would delete the ≈21 kb of SCV genome sequence that lies between the A39R and B7R-B8R loci. This would result in a virus whose propagation in SCS cells would be non-viable due to the loss of ORFs coding for essential viral proteins.

Multi-pathogen vaccines can also be generated by delivering epitopes from multiple pathogens[54,55]; however, generating antibody responses to complex folded immunogens using such an approach is intrinsically difficult. An alternative approach is mixing, such as the recently developed hexavalent childhood vaccine (DTaP5-IPV-Hib-HepB)[56]. Another mixed vaccine is the tetravalent dengue vaccine, which compromises a mixture of four chimeric viral constructs covering the four dengue serotypes[57]. A SCV-ZIKA/CHIK vaccine would (like such mixed vaccines) reduce the "shot burden" and simplify immunization schedules. In addition, the single-vector construct simplifies manufacture, ensures equal delivery of immunogen genes to antigen producing cells in vivo, and avoids formulation issues associated with mixing.

## Methods

**Ethics statement.** All mouse work was conducted in accordance with the "Australian code for the care and use of animals for scientific purposes" as defined by the National Health and Medical Research Council of Australia. Animal experiments and associated statistical treatments were reviewed and approved by the QIMR Berghofer Medical Research Institute animal ethics committee (P2195, A1604-611M) and CHIKV work was conducted in biosafety level-3 facility at the QIMR Berghofer.

**Cell lines and virus stocks.** CHO cells (ATCC CCL-61) were used to generate the SCS cell line[2]. Vero (ATCC CCL-81) and C6/36 cells (ATCC CRL1660) were purchased from the European Collection of Authenticated Cell Cultures/Sigma in 2016 and were passaged no more than 20 times. Cell lines are routinely authenticated in-house by Short Tandem Repeat profiling and checked for mycoplasma using MycoAlert™ Mycoplasma Detection Kit (Lonza). Virus stocks were checked for mycoplasma as described[58]. Low endotoxin contamination status of fetal calf serum was confirmed as described[59].

**Construction of SCV-ZIKA/CHIK.** The construction of VACV-CHIK from the Copenhagen strain of vaccinia (gifted by Dr Robert Drillien, Institute of Virology, Strasbourg, France) has been described previously[2]. SCV-ZIKA/CHIK was constructed by replacing the *B7R-B8R* ORF with a poxvirus expression cassette for ZIKV prME (Brazilian isolate ZikaSPH2015, Genbank: KU321639) and deleting the *D13L* ORF by homologous recombination (Fig. 1a). For insertion of the ZIKV prME expression cassette (by replacing the *B7R-B8R* ORFs), a homologous recombination cassette was constructed consisting of the following elements: (i) F1 homologous recombination targeting sequences upstream of the *B7R* gene, (ii) an expression cassette consisting of a vaccinia virus early/late promoter[53] operatively linked to a protein coding sequence for a fluorescent blue protein fusion with Zeocin resistance protein (BFPzeo) and ending with a poxvirus early transcriptional stop sequence, (iii) a repeat of the F1 homologous recombination arm, (iv) an expression cassette consisting of a vaccinia virus early/late promoter operatively linked to a protein coding sequence for ZIKV prME followed by a poxvirus early transcriptional stop sequence and finally (v) F2 homologous recombination targeting sequences downstream of the *B8R* gene. The signal sequence for prME was the natural predicted sequence, which constitutes the C-terminal 18 amino acids of Capsid[60]. With the addition of the ATG start codon, the N-terminal sequence of prME becomes MGADTSVGIVGLLLTTAMA-*AEVTRR* (italics represents the beginning of pr). The nucleotide sequences of the ZIKA and CHIK immunogens and insertion sites, and the *D13L* deletion (see below) are provided in Supplementary Fig. 3.

The *D13L* ORF was replaced by *DsRed/CP77*[61] (Fig. 1a) as described in the construction of SCV-CHIK[2]. Homologous recombination was performed by transfecting both the ZIKV prME and *D13L* homologous recombination cassettes into CHO+D13L cells[2] previously infected with VACV-CHIK at an MOI of 0.01 PFU/cell. SCV$^{DsRed/CP77}$-ZIKA$^{BFPzeo}$/CHIK was enriched from the homologous recombination infection by amplifying the virus in CHO+D13L cells in the presence of Zeocin (ThermoFisher Scientific) followed by a second infection of a fresh set of CHO+D13L cells in the presence of Zeocin with amplified virus. These infected cells were recovered and made into a single cell suspension by TrypLE Select digestion and then single cell sorted so that a single blue and red fluorescent cell was seeded into one well of a 96-well plate containing CHO+D13L cells using FACSAria Fusion flow cytometer (BD Biosciences). After incubation in the presence of Zeocin, wells containing a single blue and red fluorescent focus of infection were harvested and resuspended as single cell suspensions before single cell sorting and seeding into 96-well plates of fresh CHO+D13L cells. This single cell sorting and culturing in the presence of Zeocin was repeated five times in order to eliminated trace contamination with the original VACV-CHIK and produce clonal SCV$^{DsRed/CP77}$-ZIKA$^{BFPzeo}$/CHIK. A number of clonally purified SCV$^{DsRed/CP77}$-ZIKA$^{BFPzeo}$/CHIK vectors were amplified in SCS cells in the absence of Zeocin and were then subject to PCR analysis to confirm insertion of ZIKV prME into the *B7R-B8R* locus, retention of the CHIK expression cassette in the *A39R* locus, removal of *D13L* and absence of contaminating VACV-CHIK. Clones were amplified in SCS cells in the absence of Zeocin to encourage intramolecular recombination between the F1 homologous recombination sequence and the F1 repeat sequence resulting in the deletion of both *BFPzeo* and *DsRed/CP77* expression cassettes. Single cell suspensions of these cultures were bulk sorted (FACSAria) and non-fluorescent cells retained. The PCR was repeated to confirm retention of inserts, removal of *D13L*, and loss of *BFPzeo* and *DsRed/CP77*. All SCV vaccine stocks were prepared in SCS cells and titred as described[2].

**PCR of SCV-ZIKA/CHIK-infected SCS cells.** Construction and purity of SCV-ZIKA/CHIK was confirmed by PCR. SCS cells were infected with SCV-ZIKA/CHIK (MOI = 1) and after 24 h DNA was extracted using a DNAeasy kit (Qiagen). PCR reactions were performed using KAPA HiFi polymerase (Kapa Biosystems) with the following primer pairs: *A39R* locus, forward 5′-GTCGTA-CAATTCTGTACCTATCAAGG-3′ and reverse 5′-CGCATCTGTATCAAACG-GAGG-3′; *B7R-B8R* locus, forward 5′-GGTGCTTCGTACATAAGTTGT-3′ and reverse 5′-GGAATCACTATTACTACTTGT-3′; and *D13L* locus, forward 5′-CGACACCCGTTTCATGGAACAA-3′ and reverse 5′-GGACGACGAGA-TACGTAGAGTGT-3′. PCR products were run on a 1% agarose gel and visualized using GelRed (Biotium). Uncropped images are provided in Supplementary Fig. 4.

**Western blotting of SCV-ZIKA/CHIK-expressed immunogens.** Western blotting of CHIKV antigens in lysates of SCV-ZIKA/CHIK-infected SCS cells (24 h, MOI = 1) and HeLa cells (MOI = 5) at the indicated time(s) was undertaken as described previously using anti-CHIKV polyclonal anti-sera[2] (1 in 100 dilution) generated in-house by immunizing C57BL/6 mice twice with inactivated CHIKV. The lysates were also analyzed using an anti-ZIKV E protein mouse polyclonal serum, generated in-house by immunization with *E. coli*-derived purified recombinant ZIKV E protein formulated with Titremax Gold adjuvant (Sigma-Aldrich). Uncropped images are provided in Supplementary Fig. 4.

**Assessment of SCV-ZIKA/CHIK insert stability.** SCV-ZIKA/CHIK was passaged ten times in subconfluent SCS cells, with 3 days of culture for each passage. After each culture period, cells were collected by scraping and were homogenized (as described[2]) and reseeded at MOI = 0.01–0.001. The MOI was estimated by plaque formation in SCS cells. PCR of infected cells after passage 1 and passage 10 was performed for the *A39R* and *B7R-B8R* loci on cell lysates as described above. For quantitative PCR, DNA was extracted from lysates using the Magery Nagel Nucleospin tissue extraction kit (Scientifix, Vic, Australia). ZIKV M primers were F 3′ TTGGTCATGATACTGCTGATTGC 5′, R 3′ CCTTCCACAAAGTCCCTA-TTGC 5′ and VACV *G1L* primers were F 3′ TCGGTGTCTATAACGGAAC 5′, R 3′ GTTTAGTCGTGTCTACAAAAGG 5′. Quantitative PCR was undertaken (in duplicates) using the CFX 96 touch PCR detection system (Biorad) using the same cycling parameters for both sets of primers; 1 × 50 °C 2 min, 1 × 95 °C 2 min, 45 × 94 °C 5 s, 52 °C 10 s and 72 °C 40 s), with analysis using Biorad CFX Real Time Analysis software.

**Vaccination and antibody responses.** Groups of male or female (as indicated) 6–8-week-old C57BL/6J (ARC, Canning Vale, WA, Australia) or IFNAR$^{-/-}$ mice[27] were vaccinated intramuscularly (50 μl into both quadriceps femoris muscles) with SCV-CHIK, SCV-ZIKA/CHIK or SCV-cont (a control SCV vaccine encoding dsRed[2]) in 10 mM Tris HCl pH 8. Serum antibody responses were determined by standard ELISA using (i) whole inactivated CHIKV as antigen as described[62] and (ii) whole-ZIKV$_{766}$ virus preparations purified from infected Vero cell supernatants by 8% polyethylene glycol precipitation and ultracentrifugation through a 20% sucrose cushion. CHIKV neutralization titers against the Reunion Island isolate (LR2006-OPY1) of CHIKV were determined as described[63]. As ZIKV$_{Natal}$ plaques poorly, ZIKV neutralizing antibody titers were determined (against ZIKV$_{Natal}$ and ZIKV$_{MR766}$) by incubating dilutions of heat-inactivated mouse serum (in duplicate) with 100 CCID$_{50}$ of virus for 3 h before adding Vero E6 cells (Sigma, ECACC Vero C1008) (10$^4$/well of a 96-well plate). After 5 days cells were fixed, stained with crystal violet and the reciprocal 50% neutralizing titers determined.

**CHIKV and ZIKV challenge.** Vaccinated C57BL/6 mice were challenged with 10$^4$ CCID$_{50}$ of the Reunion Island CHIKV isolate subcutaneously into the feet, and the post-challenge viremia and foot swelling determined as described[4,63]. Vaccinated IFNAR$^{-/-}$ mice were challenged by s.c. infection with 10$^3$ CCID$_{50}$ of ZIKV$_{MR766}$ or 10$^4$ CCID$_{50}$ ZIKV$_{Natal}$ and viremia assessed as described[27]. ZIKV$_{MR766}$ infected female IFNAR$^{-/-}$ mice were killed when ethically defined end points were reached (primarily hind-limb paralysis). ZIKV$_{Natal}$ infection of pregnant IFNAR$^{-/-}$ dams was undertaken at E6.5 or E12.5 with viremia in dams measured as described[27]. At E17.5 dams were euthanized, fetuses weighted and photographed, and fetal heads and placenta were either (i) processed for determination of tissue titers[27] or (ii) placed into RNAlater (Ambion, Austin TX, USA) and processed for qRT PCR[63] using ZIKV M primers (as described above), with normalization to RPL13A[27].

Male IFNAR$^{-/-}$ mice were infected with ZIKV$_{Natal}$ and viremia determined as described[27]. On day 21 post infection testes were removed and analyzed for qRT PCR as above, and H&E staining as described[3]. Immunohistochemistry was undertaken using the 4G4 monoclonal antibody generated in-house (and used as neat hybridoma supernatant) as described[27,64] and Warp Red Chromogen kit (Biocare Medical, CA, USA).

**Prior infection with VACV, CHIKV or SCV-CHIK.** Female 6–8-week-old C57BL/6 mice were infected with VACV (replication competent Copenhagen strain of vaccine) 10$^6$ pfu i.p. or with CHIKV (LR2006-OPY1) 10$^4$ CCID$_{50}$ s.c. in the feet[4] or with SCV-CHIK 10$^6$ pfu i.m.

**Statistics.** Statistical analysis of experimental data was performed using IBM SPSS Statistics for Windows, Version 19.0. Two-sample comparison using $t$ test was performed when the difference in variances was <4, skewness was ≥2 and kurtosis was <2. Non-parametric data with difference in variances of <4 was analyzed using Mann–Whitney $U$ test, if difference of variances was >4 the Kolmogorov–Smirnov test was employed. The log rank (Mantel–Cox) test was used for statistical analysis of surviving proportions.

**Data availability.** The authors declare that the data supporting the findings of this study are available with the article and its Supplementary Information files, or are available from the authors upon request.

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

## Acknowledgements
We thank the QIMR B animal house staff for their assistance and supply of GMO mice, and Clay Winterford and his team at QIMR B for their help with histology and immunohistochemistry. We thank Dr I Anraku for his assistance in managing the BSL3 facility at QIMR B. Thanks also to Caitlin O'Brien (UQ) for help with the ZIKV antigen preparations. N.A.P. was supported by an Advance Queensland Research Fellowship from the Queensland Government, Australia. A.S. holds a Principal Research Fellowship from the National Health and Medical Research Council of Australia. This work was also supported in part by Sementis Ltd. via (i) Australian Department of Industry, Enterprise Connect Researchers in Business Fellowships for L.L. and T.H.C., (ii) a Science Industry Endowment Fund, STEM+ Business Fellowship for T.H.C and L.L. and (iii) an Australian Research Council Linkage grant LP160100633 awarded to J.D.H. and P.M.H. E.N. was supported in part by the Daiichi Sankyo Foundation of Life Science.

## Author contributions
P.M.H. designed the vaccines, directed the project, and commissioned the work. L.L., P. E., and T.H.C. constructed and characterized the vaccines, supervised by K.R.D. and J.D. H. N.A.P., E.N., T.T.L., K.Y., J.H., and B.T. undertook the animal experiments. N.A.P. and A.S. designed and supervised the animal experiments and analyzed the results. Y.X.S, A.A.K., and J. H.-P. provided vital characterized reagents and assay support. A.S. wrote the manuscript.

## Additional information

**Competing interests:** This research was supported in part by funding from Sementis Ltd. in which P.M.H. and J.D.H. are shareholders. A.S. is an unpaid member of the Sementis Ltd. Scientific Advisory Board and has undertaken contract R & D for Sementis Ltd. Part of the work described in this manuscript has been filed in a patent application PCT/AU2017/050879. The remaining authors declare no competing interests.

