## [Peer Review File(PDF 187 kb) · Nature Communications]

Reviewers' comments:

Reviewer #1 (Remarks to the Author):

Review on NCOMMS-17-25696 Prow et al.

This manuscript reports from the characterization of a novel vector vaccine based on recombinant vaccinia virus (VACV) and co-expressing the major structural antigens of both chikungunya virus (CHIKV) and Zika virus (ZIKV). The new vector vaccine is based on an elegant VACV technology platform which allows to generate and to produce replication-deficient but (fully) expression-competent recombinant viruses. This novel platform technology could offer major advantages with regard to the safety and efficiency of candidate vaccines similar to those shown with the safety-tested VACV MVA vaccine platform technology. Several recombinant VACV vaccines against CHIKV have been described (García-Arriaza J, *J Virol* 2014,88:3527; Weger-Lucarelli J, *PLoS Negl Trop Dis* 2014, 24:e2970; van den Doel P, *PLoS Negl Trop Dis* 2014, 8:e3101; Weber *PLoS Negl Trop Dis* 2015, 9:e0003684; and ref2). Yet, to the best knowledge of this reviewer this is the first description of a VACV candidate vaccine successfully targeting both CHIKV and ZIKV. This is an exciting and technically very well-done study. Statistical analyses have been appropriately done. This work firstly demonstrates the possibility to develop a single VACV-based vaccine providing protection against these important and potentially co-circulating arboviruses. The following points could be addressed.

Specific comments:

1. Results / design: For description of the B8R deletion the first publication Verardi 2001 *JVI* 75:11 should be referenced instead of ref 27.

2. Results Fig. 1c / PCR analysis of the viral genome: The recombinant virus SCV-ZIKA/CHIK is a quite complex construct using strong synthetic VACV promoters for transcriptional control the recombinant gene expression. To ensure that this virus could indeed be a viable candidate vaccine it is absolutely essential to demonstrate its genetic stability. This can be quite easily done by a somewhat more complete PCR monitoring of the recombinant gene sequences in the genomic insertion sites following serial passages (e.g. 5-9 low MOI amplification cycles mimicking a production process) in the producer SCS cell line.

3. Results Fig. 1d,e / Western blot SCV-ZIKA/CHIK: The monitoring for synthesis of recombinant proteins is done on SCS cells. This cell culture allows productive replication of the viruses. Another (human derived) cell line should be more suitable to better demonstrate the expression capacity of this non-replicating vector vaccine.

4. Results Fig 1e / Western blot SCV-ZIKA/CHIK: It would be useful to include SCV-CHIK in this analysis and to demonstrate the synthesis of equivalent amounts of recombinant CHIKV antigens for both viruses. This can be done most convincingly by monitoring several time

points after in vitro infection.

5. Results Fig 2/3: anti-CHIKV response: SCV-ZIKA/CHIK seems to consistently induce higher anti-CHIKV antibody responses compared to SCV-CHIK. This is somewhat surprising seen that the identical expression cassette is used in both recombinant viruses. This observation needs an explanation.

6. Discussion/ In 260 ff: It should be appropriate to reference previous – yet less complex - VACV vector vaccines allowing for immunization against multiple pathogens viruses (Paoletti et al. PNAS 1984 81:193; Perkus et al Science 1985 229:981).

Reviewer #2 (Remarks to the Author):

In this manuscript, the authors construct a vaccinia vector system that expresses two viral protein antigens capable of eliciting an immune response against the medically important pathogens Zika virus (ZIKV) and chikungunya virus (CHIKV).

The vaccinia virus vector strategies employed here are not conceptually new. Vaccinia vector systems that express multiple non-viral genes have been described previously, as has the use of complementing cell lines. The SCV “platform” nomenclature throughout advertisement for a commercialized version of some of the concepts demonstrated with the WR vaccinia strain was unappealing. That the vaccinia vector expressing CHIKV structural proteins was described previously by some of these authors reduces novelty here, as they authors have simply added an additional antigen. That said, there is certainly considerable translational potential here that should simulate interest by the field.

The paper is well written and I found the experiments to be compelling. Here the authors have performed many of the same studies use to characterize CHIKV or ZIKV vaccines in different manuscripts. This is certainly not the smallest publishable unit. I do not have experimental suggestions related to points that are required to support the authors conclusions.

Overall, enthusiasm is high with respect to this journal. Nice work.

Minor points:

I appreciate that the authors made an effort to carefully describe the statistical analysis of their data (which test etc.). This made reading the paper easier.

There are many ZIKV prME vaccine constructs being developed that employ different platforms. While many of these appear promising in mine and non-human primates, it is not yet clear if they will be clinically useful. One subtle difference among many of these constructs is the signal sequence used at the amino-terminus of prM. To facilitate

comparisons among the antigens in all the vectors being evaluated as vaccines, providing the identity of the signal sequence (presumably the ZIKV one) and starting 4-5 amino acids would be useful.

The authors indicate the limit of detection in their IgG ELISA assay is "1 in 30" (e.g. Figure 2b). This may be confusing. Is this mean to be the equivalent of a 1 to 30 dilution of sera? If so, some changes to the text would be helpful.

RESPONSES TO REVIEWERS' COMMENTS

The reviewers' comments are reproduced in their entirety in italics

Reviewer #1 (Remarks to the Author):

This manuscript reports from the characterization of a novel vector vaccine based on recombinant vaccinia virus (VACV) and co-expressing the major structural antigens of both chikungunya virus (CHIKV) and Zika virus (ZIKV). The new vector vaccine is based on an elegant VACV technology platform which allows to generate and to produce replication-deficient but (fully) expression-competent recombinant viruses. This novel platform technology could offer major advantages with regard to the safety and efficiency of candidate vaccines similar to those shown with the safety-tested VACV MVA vaccine platform technology. Several recombinant VACV vaccines against CHIKV have been described (García-Arriaza J, J Virol 2014,88:3527; Weger-Lucarelli J, PLoS Negl Trop Dis 2014, 24:e2970; van den Doel P, PLoS Negl Trop Dis 2014, 8:e3101; Weber PLoS Negl Trop Dis 2015, 9:e0003684; and ref2). Yet, to the best knowledge of this reviewer this is the first description of a VACV candidate vaccine successfully targeting both CHIKV and ZIKV. This is an exciting and technically very well-done study. Statistical analyses have been appropriately done. This work firstly demonstrates the possibility to develop a single VACV-based vaccine providing protection against these important and potentially co-circulating arboviruses. The following points could be addressed.

Specific comments:

1. Results / design: For description of the B8R deletion the first publication Verardi 2001 JVI 75:11 should be referenced instead of ref 27.

This reference has been changed as requested.

2. Results Fig. 1c / PCR analysis of the viral genome: The recombinant virus SCV-ZIKA/CHIK is a quite complex construct using strong synthetic VACV promoters for transcriptional control the recombinant gene expression. To ensure that this virus could indeed be a viable candidate vaccine it is absolutely essential to demonstrate its genetic stability. This can be quite easily done by a somewhat more complete PCR monitoring of the recombinant gene sequences in the genomic insertion sites following serial passages (e.g. 5-9 low MOI amplification cycles mimicking a production process) in the producer SCS cell line.

We have serially passaged SCV-ZIKA/CHIK 10 times in SCS cells as requested (low MOI, 3 day passages). We have then used PCR and quantitative PCR to illustrate that gene inserts sizes remained the same (new Fig. 1f) and the level of insert by qPCR was undiminished (Fig. 1g). These results attest to the stability of the construct. We have added an extra Results section and a line to the discussion to cover these findings.

Instability of inserts in some MVA construct may have triggered this reviewer's question, and our current understanding is that these may have arisen from placing recombinant inserts into a deleted region of the MVA genome. The deletions occurred during the large number of passages that created MVA from VACV (Meyer), and may have arisen from homologous recombination. Thus placing an insert into a region already prone to deletion may have been a less than optimal strategy. SCV only has one deletion D13L so this potential issue does not arise.

3. Results Fig. 1d,e / Western blot SCV-ZIKA/CHIK: The monitoring for synthesis of recombinant proteins is done on SCS cells. This cell culture allows productive replication of

the viruses. Another (human derived) cell line should be more suitable to better demonstrate the expression capacity of this non-replicating vector vaccine.

Two extra panels have been added to Fig 1; Fig. 1d showing ZIKA E expression in HeLa cells, and Fig 1e showing CHIK structural protein expression in HeLa cells, both following SCV-ZIKA/CHIK infection.

4. Results Fig 1e / Western blot SCV-ZIKA/CHIK: It would be useful to include SCV-CHIK in this analysis and to demonstrate the synthesis of equivalent amounts of recombinant CHIKV antigens for both viruses. This can be done most convincingly by monitoring several time points after in vitro infection.

This data has been added as a new Fig. S1 and shows that CHIKV immunogen expression in SCV-ZIKA/CHIK infected HeLa cells was not lower than CHIKV immunogen expression in SCV-CHIK infected cells.

5. Results Fig 2/3: anti-CHIKV response: SCV-ZIKA/CHIK seems to consistently induce higher anti-CHIKV antibody responses compared to SCV-CHIK. This is somewhat surprising seen that the identical expression cassette is used in both recombinant viruses. This observation needs an explanation.

This is referring to Fig 2b and d, and 3b and d. We also noticed this apparent trend, but this did not reach significance in multiple experiments (p ranging from 0.06 to 0.3). (Note a lot of this data is non-parametric). The exception is Fig 2b (p=0.031); however, we have undertaken a number of repeat experiments contemplating some kind of “cross adjuvanting” activity, but these did not reach significance. At this stage we have no compelling evidence that any significant effect does indeed exist and are thus somewhat reluctant to speculate.

We have added a line to the relevant figure legends stating “Differences in anti-CHIKV titers between SCV-CHIK and SCV-ZIKA/CHIK were not significant” to make our current findings clear. We have also added a small paragraph to the discussion on this issue and have added a new Supplementary Fig. S2, which shows that anti-ZIKV response were not also significantly different after SCV-ZIKA/CHIK and SCV-ZIKA vaccination.

6. Discussion/ In 260 ff: It should be appropriate to reference previous – yet less complex - VACV vector vaccines allowing for immunization against multiple pathogens viruses (Paoletti et al. PNAS 1984 81:193; Perkus et al Science 1985 229:981).

Indeed a very pertinent addition; although the first reference does not actually provide any data on multi-pathogen vaccines but simply states what the group are going to do next, with the second reference providing that data. We have added an extra sentence to the discussion to describe the findings from this second reference; the Perkus paper used 2 rabbits to show responses to three different viral surface antigens after vaccination with a vaccinia virus construct encoding the 3 immunogens.

Reviewer #2 (Remarks to the Author):

In this manuscript, the authors construct a vaccinia vector system that expresses two viral protein antigens capable of eliciting an immune response against the medically important pathogens Zika virus (ZIKV) and chikungunya virus (CHIKV).

The vaccinia virus vector strategies employed here are not conceptually new. Vaccinia vector systems that express multiple non-viral genes have been described previously, as has the use of complementing cell lines. The SCV “platform” nomenclature throughout advertisement for a commercialized version of some of the concepts demonstrated with the WR vaccinia strain

was unappealing. That the vaccinia vector expressing CHIKV structural proteins was described previously by some of these authors reduces novelty here, as they authors have simply added an additional antigen. That said, there is certainly considerable translational potential here that should simulate interest by the field.

The paper is well written and I found the experiments to be compelling. Here the authors have performed many of the same studies use to characterize CHIKV or ZIKV vaccines in different manuscripts. This is certainly not the smallest publishable unit. I do not have experimental suggestions related to points that are required to support the authors conclusions.

Overall, enthusiasm is high with respect to this journal. Nice work.

Minor points:

I appreciate that the authors made an effort to carefully describe the statistical analysis of their data (which test etc.). This made reading the paper easier.

There are many ZIKV prME vaccine constructs being developed that employ different platforms. While many of these appear promising in mine and non-human primates, it is not yet clear if they will be clinically useful. One subtle difference among many of these constructs is the signal sequence used at the amino-terminus of prM. To facilitate comparisons among the antigens in all the vectors being evaluated as vaccines, providing the identity of the signal sequence (presumably the ZIKV one) and starting 4-5 amino acids would be useful.

The following details have been added to the Methods (Construction of SCV-ZIKA/CHIK); “The signal sequence for PrME was the natural predicted sequence, which constitutes the C-terminal 18 amino acids of Capsid⁵⁸. With the addition of the ATG start codon, the N-terminal sequence of PrME becomes MGADTSVGIVGLLLTTAMA-AEITRR (italics represents the beginning of Pr).”

The authors indicate the limit of detection in their IgG ELISA assay is “1 in 30” (e.g. Figure 2b). This may be confusing. Is this mean to be the equivalent of a 1 to 30 dilution of sera? If so, some changes to the text would be helpful.

Yes this is correct, we have added clarification to Fig 2 legend: The limit of detection was 1 in 30, meaning that a 1 in 30 dilution of sera was the highest (starting) concentration of sera used in the assay.

REVIEWERS' COMMENTS:

Reviewer #1 (Remarks to the Author):

The authors have appropriately addressed all questions raised in my review. Relevant experimental data have been added and the revised manuscript has been clearly improved. I have no further comments.